# Trust Me, I'm Calibrated: Robustifying Deep Networks

## Abstract

The tremendous success of deep neural networks (DNNs) in solving a wide range of complex computer vision tasks has paved the way for their deployment in real-world applications. However, challenges arise when these models are exposed to natural adversarial corruptions that can occur in unconstrained physical environments. Such corruptions are inherently present in the real world and can significantly degrade model performance by causing incorrect predictions. This vulnerability is further enhanced by the miscalibration of modern DNNs, where models tend to output incorrect predictions with high confidence. To ensure safe and reliable deployment, it is crucial to calibrate these models correctly. While existing literature primarily focuses on calibrating DNNs, it often overlooks the impact of adversarial corruption. Thus, substantial scope remains to explore how calibration techniques interact with adversarial robustness and whether improving calibration can increase robustness to corrupted or adversarial data. In this work, we aim to address this gap by employing uncertainty quantification methods to improve the calibration and robustness of DNNs and Transformer-based models against adversarial data.

## 1 Introduction

Many deep neural networks have emerged with the remarkable success of deep learning architectures in nearly every area of computer vision. However, these networks are often miscalibrated (Guo et al., 2017; Minderer et al., 2021), as their predicted confidence does not align with the actual accuracy. For example, if a DNN predicts 'dog' with 80% confidence, it should be correct exactly 80% of the time when making such predictions. Perfect calibration is defined as:

$$P(Y = \hat{Y} \mid \hat{P} = p) = p, \quad \forall p \in [0, 1]$$

where $\hat{Y}$ is the predicted label, $Y$ is the true label, and $\hat{P}$ is the model's predicted confidence.

It makes it challenging to deploy DNNs in real-world scenarios, where they must provide reliable estimates of their uncertainty. One of the primary causes of miscalibration in deep neural networks (DNNs) is the training paradigm. DNNs are typically optimized to align with one-hot encoded labels, which makes the model assign maximum confidence to the predicted class. Even when the model achieves perfect classification on the training set, the negative log-likelihood (NLL) can be further minimized by increasing the confidence of its predictions. This tendency leads to overfitting on the NLL objective, resulting in overconfident predictions and calibration errors (Guo et al., 2017). Such miscalibration poses significant challenges when deep neural networks are evaluated on corrupted data—overall accuracy drops even though the models maintain high confidence in their predictions. This issue is particularly critical, as models become vulnerable to natural corruptions that mimic the effects of adversarial perturbations (Guo et al., 2020; Agarwal et al., 2020b; Hendrycks & Dietterich, 2019a). The seriousness of this vulnerability can be seen from the fact that these corruptions are inherently present in images Agarwal et al. (2020b) without the hassle of artificially generating them. Another serious barrier to deep learning model deployment in real-world applications is incorrectly calibrating their prediction confidence. In practical scenarios, most models reflect overconfidence in their prediction probabilities even when the model predictions are wrong (Lakshminarayanan et al., 2017; Wei et al., 2022; Minderer et al., 2021).

Current research on CNN robustness primarily focuses on two areas: improving the model's robustness

to adversarial attacks (Zhang et al., 2023; Peng et al., 2023) and distinguishing between authentic and adversarial images Sen et al. (2023); Agarwal et al. (2023). They have not explored much to improve the robustness using calibration methods. We are not undermining the effort put in developing these defenses, such as binary classifiers, which are even generalized in handling unseen perturbations (Agarwal et al., 2021; 2020a) and adversarial training (Qian et al., 2022) which re-train the model using the adversarial images. However, in both these compelling defense cases, several issues involved: (i) training of a separate classifier, (ii) computational cost in generating adversarial examples, and (iii) trade-off between robustness and clean accuracy. We aim to demonstrate that effective calibration techniques can significantly enhance the robustness of deep neural networks (DNNs) and transformers against adversarial inputs. By ensuring that model confidence levels more accurately reflect the actual likelihood of correct predictions, these techniques help the network to increase the robustness against adversarial data. **Henceforth, this research aims to tackle several critical bottlenecks in the existing work: a limited exploration of defense against natural corruption, avoiding training extra classifiers or generation of adversarial examples, and no existing study understanding the correlation between confidence and robustness.** Recent literature addresses calibration from multiple perspectives: regularization-based methods (Lin et al., 2017; Szegedy et al., 2016) that improve calibration during training, post-hoc techniques (Guo et al., 2017; Kull et al., 2017) that adjust predictions after training, and uncertainty-based methods leveraging Bayesian principles to quantify model confidence. This paper focuses on Bayesian methods for calibrating deep neural networks (DNNs) to improve robustness against natural corruptions and adversarial attacks. By doing so, we aim to fill an essential gap in the literature, exploring how uncertainty-based calibration can help models better withstand adversarial challenges. Our findings reveal that CNN and transformer models often become overconfident when predicting corrupted data, increasing their vulnerability to adversarial perturbations. The Bayesian method effectively addresses natural corruption while avoiding the additional computational overhead typically associated with adversarial training. For that, extensive experiments are performed using multiple benchmark object recognition datasets, namely CIFAR- 10 (Alex, 2009), CIFAR-100 (Alex, 2009), ImageNet (Deng et al., 2009) and classification networks such as VGG (Simonyan & Zisserman, 2014) PreActResNet (He et al., 2016), Vision Transformer (ViT) (Dosovitskiy et al., 2020). These models are trained to capture the uncertainty within the model effectively. The primary contributions of this research are:

1. Conducted a comprehensive investigation into how uncertainty quantification methods affect the calibration of deep neural networks and transformers.

2. Identifying overconfident predictions as a key factor contributing to reduced robustness in CNN architectures such as VGG, ResNet, and Vision Transformers, particularly against adversarial and corrupted inputs.

3. Applied SWAG (Maddox et al., 2019b) and introduced a modified variant of SWAG with batch means, which improved robustness to distributional shifts and image corruptions.

## 2 Related Work

**Calibration Methods** Various approaches have been proposed to improve the calibration of deep neural networks (DNNs). These methods can broadly be categorized into train-time, post-hoc calibration, and data augmentation strategies. **Train-time Methods** are applied during training to modify the loss function to mitigate overconfidence. For example, Focal Loss (Lin et al., 2017) down-weights the loss contribution from well-classified examples, thereby emphasizing hard-to-classify samples. This dynamic focus inherently reduces overconfidence and improves calibration. Similarly, label smoothing (Szegedy et al., 2016) regularizes the network by distributing a portion of the target probability mass from the actual class to other classes, thus preventing the network from becoming excessively confident. Instead of assigning a probability of 1 to the correct class and 0 to all others, Label Smoothing distributes some probability mass to incorrect classes, reducing the model's overconfidence. Recent work, such as that by Park et al. (2023), further reinforces the benefits of such approaches. **Post-hoc Calibration** are methods that adjust the model's outputs after training. Temperature scaling, as demonstrated by (Guo et al., 2017), is an effective post-hoc calibration method that adjusts the softmax logits by a scalar temperature parameter to achieve better-

calibrated predictions. Building on this idea, subsequent studies have explored variations such as enhanced temperature scaling Kull et al. (2017) and meta-gradient learning approaches Bohdal et al. (2021). Other post-hoc methods include class distribution-free strategies, as introduced in Islam et al. (2021), aiming to calibrate DNNs without relying on explicit class information. **Data Augmentation Techniques** methods have also been used to improve calibration. Mixup, introduced by Thulasidasan et al. (2019), generates new training examples by combining existing ones, which helps reduce overconfidence by smoothing the decision boundaries. Additionally, hyperparameter ensembles, as proposed by Wenzel et al. (2020), leverage multiple models with varied hyperparameters to enhance calibration through ensemble averaging. Together, these methods provide a comprehensive toolkit for addressing the calibration challenges in DNNs, each offering unique advantages depending on the specific application and desired outcomes.

**Image Corruptions:** Several studies have explored the susceptibility of CNNs to common corruption. (Guo et al., 2020) shows that motion blur, commonly occurring in real-world scenarios, can significantly degrade deep learning model performance. Additionally, Agarwal et al. (2020b) introduces camera-inspired perturbations, simulating noise from natural conditions and camera imperfections to study their impact on model robustness. Similarly, Özdenizci & Legenstein (2023) focuses on addressing environmental noises like snow introduced by adverse weather conditions using diffusion methods. Dodge & Karam (2016), show that CNNs are particularly vulnerable to blur and Gaussian noise. To evaluate the robustness of neural network models, corrupted versions of standard datasets have been widely used, as proposed by Hendrycks & Dietterich (2019b). These datasets introduce various noise and distortions, categorized systematically into different classes.

**Improving Robustness against corruptions:** Image restoration and enhancing model robustness against various corruptions have been the focus of many studies. For instance, Cui et al. (2023) introduces a multi-scale representation to improve image quality by addressing blur levels and noise in corrupted images. Dong et al. (2023) focuses on utilizing multi-scale processing to remove motion blur through residual learning and low-pass filters, offering a comprehensive approach to handling complex distortions. Cheng et al. (2024) proposes a novel denoising method using a truncated loss function within a Res2Net architecture. This technique efficiently suppresses non-Gaussian noise, including impulse noise like shot noise, while preserving crucial image details and edges. Furthermore, Zhu et al. (2023) introduces a method that restores images degraded by weather conditions, such as snow and fog. The approach learns weather-general features standard across different adverse weather types and weather-specific features unique to individual conditions, enhancing the model's adaptability to diverse environmental distortions. Researchers are also exploring whether there is any connection between corruption and adversarial perturbation that can be employed for a universal defense (Agarwal et al., 2022b;a).

## 3 Uncertainty Quantification Methods

This section outlines various approaches to uncertainty quantification within the Bayesian framework. We demonstrate that incorporating Bayesian methods to model uncertainty leads to significantly better-calibrated and more reliable predictions. Prior work has explored a range of Bayesian techniques, including confidence calibration methods (Mehrtash et al., 2020; Lakshminarayanan et al., 2017) and Bayesian Neural Networks (BNNs) (Izmailov et al., 2020). Bayesian approaches such as variational inference (Choi et al., 2019; Amini et al., 2018; Loquercio et al., 2020), sampling-based methods (Mitros & Mac Namee, 2019; Ovadia et al., 2019), and the Laplace approximation (Ritter et al., 2018) have been shown to significantly enhance the generalization performance of BNNs, particularly under out-of-distribution (OOD) scenarios. Building on these foundations, we employ Stochastic Weight Averaging-Gaussian (SWAG) (Maddox et al., 2019b) and introduce a novel variant based on batch mean estimation. This enhanced method improves model calibration and significantly boosts robustness to both natural corruptions and adversarial perturbations.

### 3.1 Stochastic Weight Averaging Gaussian

Stochastic Weight Averaging-Gaussian (SWAG) (Maddox et al., 2019b) is a Bayesian method approximating the posterior distribution over model parameters. It extends SWA Maddox et al. (2019a) by averaging weights along the optimization trajectory while capturing uncertainty through a low-rank Gaussian approximation of the weight distribution. While traditional SGD optimizes the neural network by converging to a single set

of weights, SWAG takes a different approach. It builds on SGD by collecting multiple weight checkpoints throughout training, averaging them to explore a **broader region** of the loss landscape. SWAG then fits a Gaussian distribution to these collected weights, effectively capturing the inherent uncertainty in the model's parameters. Suppose the model weights after epoch $i$ are $\theta_i$. Then, the SWA solution after $T$ epochs is given by: $\theta_{\text{SWA}} = \frac{1}{T} \sum_{i=1}^{T} \theta_i$, With SWAG (Stochastic Weight Averaging-Gaussian) Maddox et al. (2019b), a Gaussian is fitted with the SWA mean as the first moment and a low-rank diagonal covariance matrix, thus forming an approximate posterior distribution over model weights. SWAG then estimates the covariance structure around the mean. SWAG uses both a low-rank approximation and a diagonal to capture the uncertainty in the weight space. The covariance matrix combines two components: the low-rank component, which models the directions in the parameter space where weights vary the most, and the diagonal component, which accounts for variance along each parameter independently, offering a more straightforward estimate of uncertainty: $\Sigma = \frac{1}{K-1} \sum_{i=1}^{K} (\theta_i - \bar{\theta})(\theta_i - \bar{\theta})^T$. Near convergence, SGD updates can be modeled as an Ornstein-Uhlenbeck process(Stephan et al., 2017), leading to a stationary Gaussian distribution over weights.

Using this Gaussian distribution, sample several weight sets $\mathbf{w}_{\text{SWAG}}^i$. Each sampled weight represents a different model version, incorporating the variability captured during training. Finally, we perform Bayesian model averaging to combine these predictions into one final output. SWAG can provide well-calibrated uncertainty estimates for neural networks across various settings in computer vision. Notably, it achieves a higher test likelihood than other state-of-the-art approaches, such as MC Dropout (Gal & Ghahramani, 2015) and temperature scaling (Guo et al., 2017).

### 3.2   Modified SWAG for Better Calibration

We introduce SWAG with batch means (SWAGBM) [1], an enhancement to the standard SWAG approach. In conventional SWAG, model snapshots are collected at the end of each epoch after SWA is activated. In contrast, our SWAG-BM method accumulates model weights at the batch level. These weights are temporarily stored and averaged over a fixed batch size before updating the first and second moments of the weight distribution. This additional averaging step reduces the variance in the covariance estimates, leading to a smoother and more accurate posterior approximation. As a result, this approach yields models with improved calibration and enhanced robustness, particularly in environments with noisy or adversarial data. This averaging strategy is inspired by prior work (Maddox et al., 2019a; Polyak & Juditsky, 1992), demonstrating that generalization performance can be improved by averaging model weights. By aggregating multiple weight updates in batches, our method leverages this principle to achieve a more stable and accurate representation of the posterior distribution. At test time, Bayesian Model Averaging (Fragoso et al., 2018) is applied to combine predictions and get the final prediction.

## 4   Experimental Results and Analysis

This section first discusses the ingredients needed to perform the experiments, such as datasets and CNNs. We have used multiple benchmark datasets, including CIFAR-10 and CIFAR-100, and two CNNs, namely VGG and PreActResNet. The PreActResNet-164 and VGG-16 models are trained with batch normalization on both datasets for 300 epochs. The initial learning rate is set to 0.01 with a weight decay of 0.0002. Stochastic Weight Averaging (SWA) is introduced at epoch 161 to collect the model weights, using a learning rate of 0.05. We have used the pre-defined training and testing split of datasets to evaluate the models' confidence. The models are trained using two optimization techniques, namely Stochastic Gradient Descent (SGD) and Stochastic Weight Averaging Gaussian (SWAG), to reflect the impact of calibration/confidence on their classification performance. In the end, to analyze the correlation between confidence and robustness, we have used the naturally corrupted images of the test set of the datasets (Hendrycks & Dietterich, 2019a). The corrupted images of the datasets are taken from the following link[1]. For SWAGBM, we set the batch size to 10. Through experimentation with various batch sizes, we determined that a batch size of 10 provided the optimal balance between variance reduction and stability.

---

[1]https://github.com/hendrycks/robustness?tab=readme-ov-file

---

**Algorithm 1** SWAG with Batch Means Algorithm

---

**Require:** $\theta_0$: pre-trained weights, $\eta$: learning rate, $T$: number of training steps, $K$: number of columns in deviation matrix, $S$: Batch Size, $c$: frequency of storing snapshots

1: **Initialize Moments:**
2: $\bar{\theta} \leftarrow \theta_0, \bar{\theta}^2 \leftarrow \theta_0^2$
3: $\tilde{\theta} \leftarrow 0$      ▷ Batch mean accumulator
4: $m \leftarrow 0$      ▷ Batch count
5: **for** $i = 1, 2, \ldots, T$ **do**
6:      $\theta_i \leftarrow \theta_{i-1} - \eta \nabla_\theta \mathcal{L}(\theta_{i-1})$
7:      **if** $\text{MOD}(i, c) = 0$ **then**
8:          $m \leftarrow m + 1$
9:          $\tilde{\theta} \leftarrow \tilde{\theta} + \theta_i$
10:      **end if**
11:      **if** $\text{MOD}(m, S) = 0$ **then**      ▷ Batch size reached
12:          $n \leftarrow m/S$      ▷ Number of stored batches
13:          $\bar{\theta} \leftarrow \frac{n\bar{\theta} + \tilde{\theta}/S}{n+1}$      ▷ Update mean
14:          $\bar{\theta}^2 \leftarrow \frac{n\bar{\theta}^2 + (\tilde{\theta}/S)^2}{n+1}$      ▷ Update second moment
15:          $\tilde{\theta} \leftarrow 0$      ▷ Reset batch accumulator
16:          **if** $\text{NUM\_COLS}(\hat{D}) = K$ **then**
17:             $\text{REMOVE\_COL}(\hat{D}, 1)$
18:          **end if**
19:          $\text{APPEND\_COL}(\hat{D}, \tilde{\theta}/S - \bar{\theta})$
20:      **end if**
21: **end for**
22: **Return:** $\theta_{swa} = \bar{\theta}, \quad \Sigma_{diag} = \bar{\theta}^2 - \bar{\theta}^2, \quad \hat{D}$
23: **Test Bayesian Model Averaging:**
24: **for** $i \leftarrow 1, 2, \ldots, S$ **do**
25:      Draw $\theta_i \sim \mathcal{N}(\theta_{\text{swa}}, \Sigma_{\text{diag}} + \frac{\hat{D}^T \hat{D}}{2(K-1)})$      ▷ Sample weights from posterior distribution
26:      Update batch norm statistics with a new sample.
27:      $p(y^*|Data) += \frac{1}{S} p(y^*|\theta_i)$      ▷ Average predictions

---

*We also conducted experiments on larger datasets and models such as ImageNet 1k and ViT.* We took the pre-trained ViT with patch size of $16 \times 16$ and fine tuned it on CIFAR-100 with 100 epochs. The model is trained using the Adam optimizer with a learning rate of 0.01 and a weight decay of 0.002. SWA is introduced at $60^{\text{th}}$ epoch to collect the model weights. To effectively analyze the observation presented in this paper, we have used several metrics proposed by (Maddox et al., 2019b) namely (i) **confidence:** is defined as the maximum softmax output value in the model's predictions, representing the model's certainty in its output, (ii) **perfect calibration:** In an ideally calibrated model, the predicted confidence directly aligns with the true accuracy. For example, if a model predicts with 70% confidence, it should be correct 70% of the time. Perfect calibration ensures that the confidence scores are a true reflection of the model's reliability, making it a crucial aspect of trustworthy AI systems, and (iii) **reliability diagram:** We used the modified reliability diagram as introduced in (Maddox et al., 2019b) and (Guo et al., 2017) to effectively visualize how accurately the model's confidence reflects its likelihood of correctness across different types of noise and distortions. This visualization is beneficial for understanding model behavior under various conditions, such as different types of noise, distortions, or data shifts, enabling insights into where the model might overestimate or underestimate its confidence.

## 4.1 Analysis of covariance matrix of SWAG and SWAGBM

Based on our analysis of the covariance matrices for the classifier layers, the standard SWAG method produces a matrix (Figure 7) where all diagonal entries appear uniformly bright, indicating consistently elevated variance across all elements. This implies that standard SWAG estimates high individual uncertainty and strong inter-parameter correlations within the bias vector.

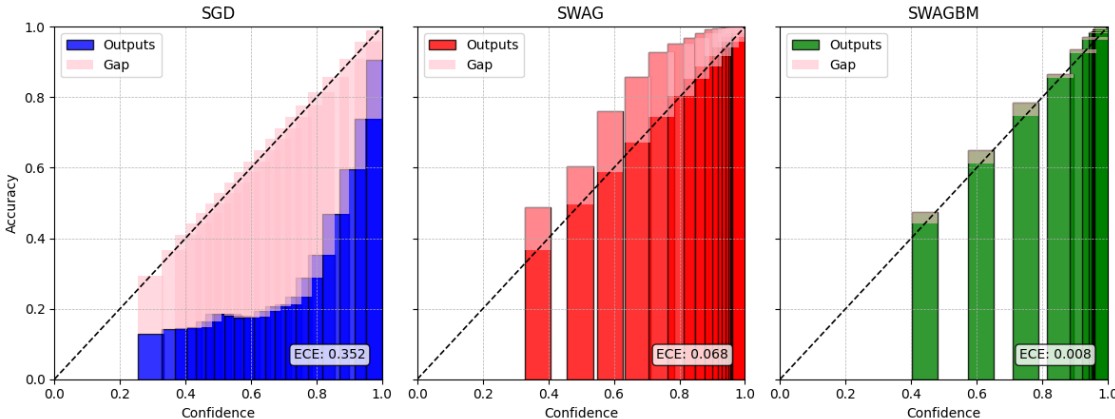

Figure 1: Reliability diagram on CIFAR10 dataset corrupted with brightness noise using VGG16BN model

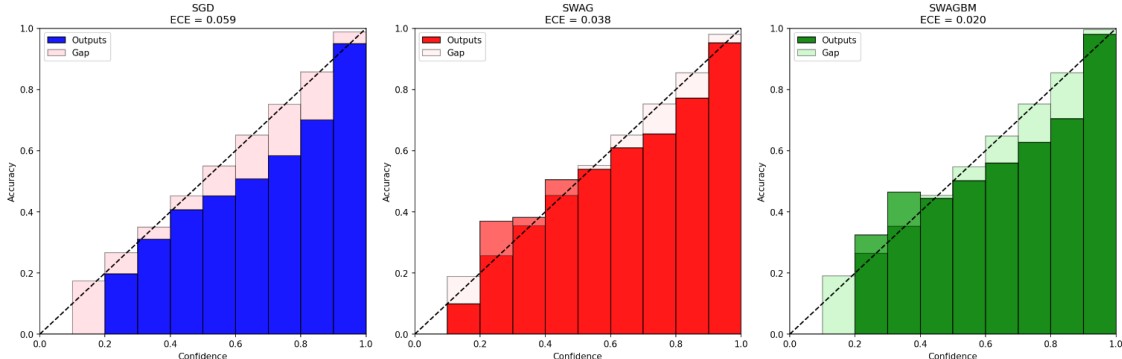

Figure 2: Reliability diagram on CIFAR100 dataset trained using ViT model.

In contrast, the SWAGBM-10 method exhibits a more moderated and uniform distribution along the diagonal, with noticeably lower intensity. This suggests that its uncertainty estimates are less extreme and more stable. Our analysis across other layers further confirms that the covariance matrices generated by SWAGBM are more stable, with reduced inter-parameter covariances.

Among the different batch sizes evaluated, SWAG with BM-10 consistently outperformed all others in delivering reliable uncertainty estimates.

## 4.2 Analysis of Calibration and Robustness

A well-calibrated model inherently provides reliable uncertainty estimates. Building on this concept, we explore the relatively underexplored calibration area in adversarial robustness. Our analysis leverages a Bayesian calibration technique that quantifies uncertainty, enhancing robustness. Additionally, SWAGBM employs weight averaging (Maddox et al., 2019a), improving the model's generalization ability. From the reliability diagrams (Figures 1 and 2), it is evident that SWAGBM and SWAG outperform SGD-trained models. While SWAG already provides improved calibration over SGD—reducing the significant overconfidence observed in SGD-trained models—SWAGBM further enhances this effect by making the predictions even more calibrated and reliable. From Figure 3 on the CIFAR-10 dataset, we observe that the VGG model trained with SGD consistently exhibits overconfident predictions when dealing with noisy or corrupted data. The prediction points are significantly above the optimal line, indicating excessive confidence. This overconfidence is evident in the sharp drop in accuracy at high confidence levels for the SGD-trained models. In contrast, the SWAG-trained models (Maddox et al., 2019b) provide more reliable uncertainty estimates, as shown by the smoother curves and higher accuracy across varying confidence levels, particularly under

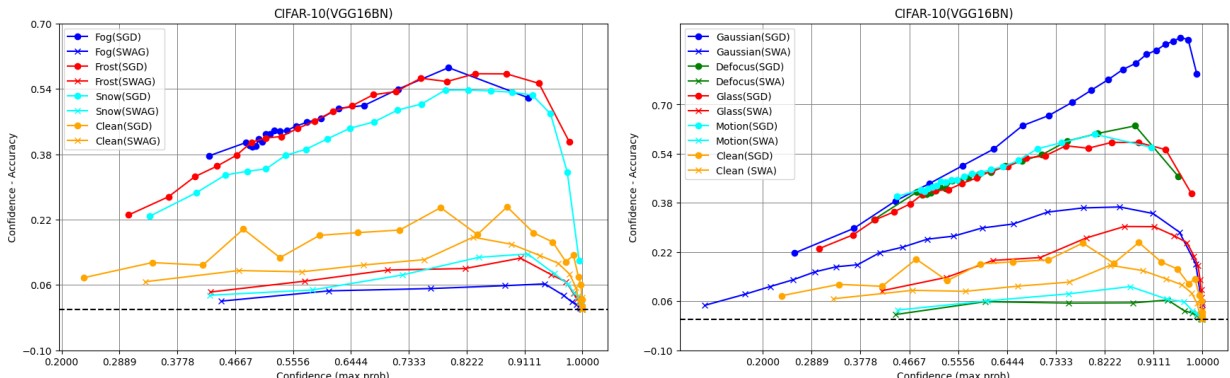

Figure 3: Reliability plots comparing the calibration of different models on CIFAR-10 images corrupted by environmental and blur distortions. The plots are reflected to showcase the calibrated capacity of the VGG model.

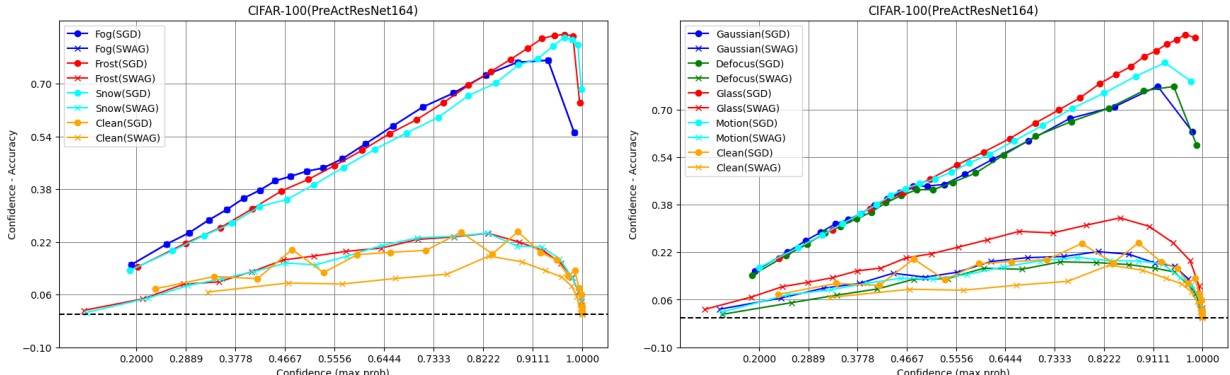

Figure 4: Reliability plots comparing the calibration of different models on CIFAR-100 images corrupted by environmental and blur distortions. The plots are reflected to showcase the calibrated capacity of the PreActResNet model.

noisy conditions. Although the SGD model also shows overconfidence in the clean dataset, it is far less pronounced than its behavior on data with different types of noise. Its effect can be seen in accuracies in Table **??**. In contrast, the predictions made using SWAG are much closer to the optimal line, demonstrating better calibration and improved performance on corrupted data. The reliability curves for the model trained with SWAG are consistently closer to the optimal line, suggesting more reliable and well-calibrated predictions across different noise types. SWAGBM and SWAG maintain more calibrated confidence levels across clean and noisy datasets. A similar observation from Figures 6 can be made on the PreActResNet, where the predictions made by the SGD model tend to be overconfident when noise is present in the data. This overconfidence is reflected in the model assigning high probabilities to its predictions, even when the input images are corrupted. Such behavior indicates that the SGD-trained model struggles to accurately quantify uncertainty in noisy conditions, potentially leading to incorrect or misleading predictions.

In the case of PreActResNet, trained on CIFAR-10, SGD achieves an accuracy of 90.27%, while SWAG improves this to 94.59%. SWAGBM enhances it to 95.4%. Similarly, on CIFAR-100, the accuracy for the clean dataset is 67.79% with SGD, but it increases to 80.37% when using SWAG. SWAG provides a more reliable method for handling various forms of corruption, thereby enhancing the performance and robustness of CNNs. Across almost all noise types, both on CIFAR-10 and CIFAR-100 datasets, the Stochastic Weight Averaging Gaussian (SWAG) models show higher accuracy than those trained with standard SGD. The most notable improvements with SWAG are seen in challenging noise conditions, such as brightness, contrast, Gaussian blur, and impulse noise, where SWAG significantly enhances performance.

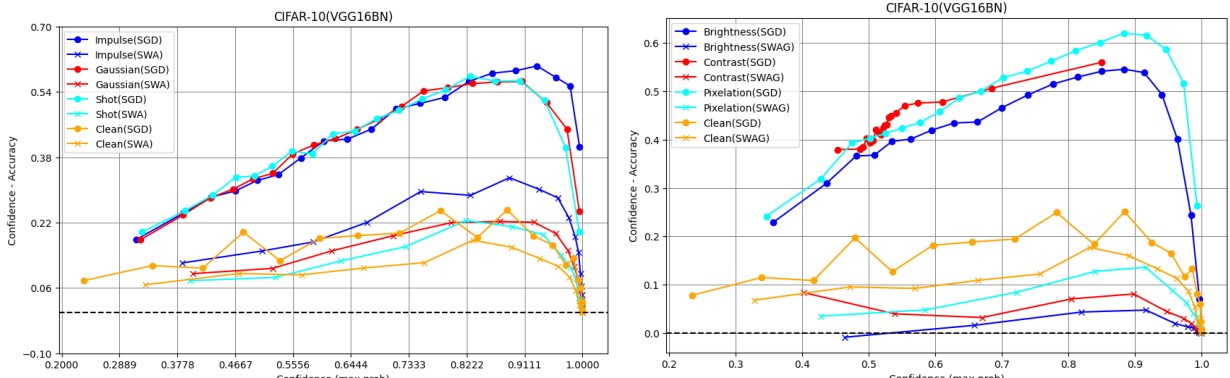

Figure 5: Reliability plots comparing the calibration of VGG-16BN on CIFAR-10 under digital and noise distortions.

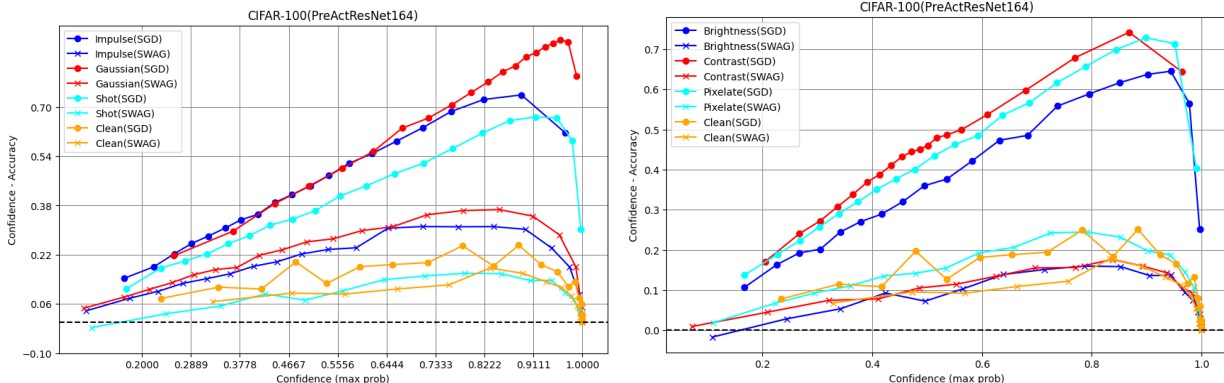

Figure 6: Reliability plots comparing the calibration of PreResNet164 on CIFAR-100 under digital and noise distortions.

### 4.3 Effect of Larger Dataset

The reliability graphs in Figure 8 illustrate model evaluations on the ImageNet dataset. The findings reveal a consistent trend: models trained with SGD demonstrate overconfidence, even with the larger dataset, while those trained with SWAG exhibit calibration more closely aligned with the ideal confidence-accuracy relationship. From Table 1, the clean ImageNet accuracy of PreActResNet-164 improves from 82% with SGD to 91% with SWAG. Under Contrast, the accuracy of the SGD-trained model drops significantly to 4.67%, whereas the SWAG model maintains a higher accuracy of 17.49%, highlighting its robustness to such perturbations. As plotted in Figure 8, the reliability diagrams show that SGD-trained models consistently show overconfident predictions. They are significantly above the optimal line. This indicates that not only on smaller datasets like CIFAR-10 and CIFAR-100 but even in more enormous datasets such as ImageNet, SGD-trained models show overconfident predictions. The SWAGBM and SWAG models improve the calibration and are more robust to corruption in larger datasets.

### 4.4 Severity Analysis

We evaluated calibration under distributional shift by plotting box plots across various severity levels. Our study compared the accuracy and Expected Calibration Error (ECE) for all corruption types on CIFAR-10 and CIFAR-100 datasets. We report the test set mean for each method and use box plots to summarize outcomes at each shift intensity level. Each box encapsulates the quartile distribution computed over all 16 shift types, with error bars marking the minimum and maximum values observed.

Figures 9, 10, and 11 display the box plots of the VGG model on the CIFAR-10 and CIFAR-100 datasets For this comparison, we employed several methods: SWA (Maddox et al., 2019a), SWAGDIA (Maddox et al.,

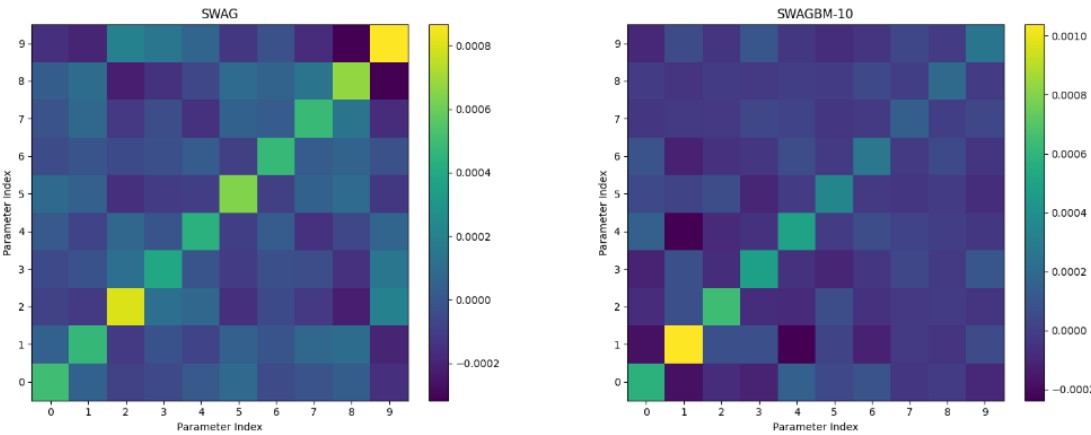

Figure 7: Covariance matrix comparison SWAG and SWAGBM

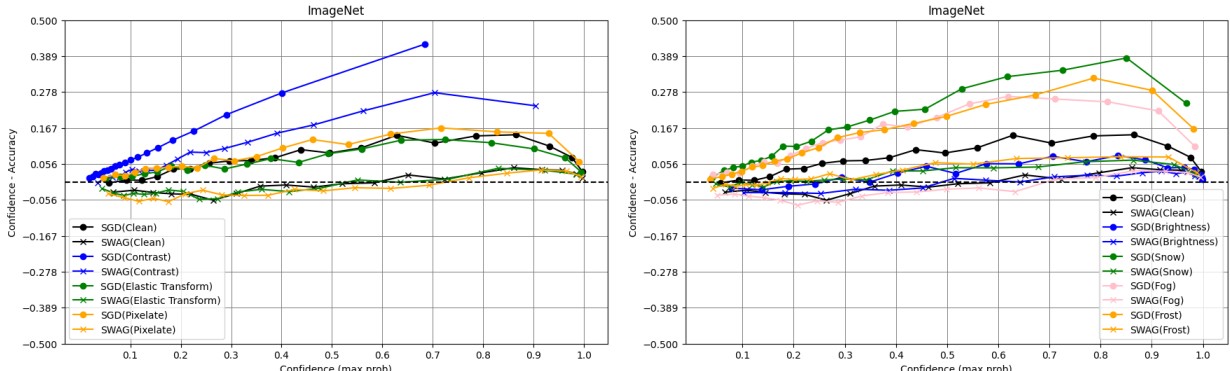

Figure 8: Reliability plots illustrating the impact of corruption on the ImageNet dataset using model Pre-ActResNet 164.

2019b), SWAG (Maddox et al., 2019b), and the KFAC Laplace (Ritter et al., 2018) model. Our proposed SWAGBM method consistently outperforms the others in every evaluation metric—accuracy, negative log-likelihood (NLL), and ECE—across different models and datasets. The improved accuracy and lower ECE and NLL values suggest that SWAGBM provides better uncertainty estimates.

## 4.5 SWAG with Vision Transformers

The calibration and robustness of Bayesian techniques for transformers have been examined in recent works, including Cinquin et al. (2021) and Chen & Li (2023). More recently, Kampen et al. investigated the effects of partially stochastic Bayesian neural networks, with a particular focus on SWAG, in the context of transformer models for NLP tasks. These insights show how to extend similar methodologies to vision transformer models, potentially enhancing their uncertainty quantification and generalization capabilities. Therefore, we have trained ViT using SWAG and SWAGBM methods and observed an increase in accuracy from 97.02% to 98.9%, as indicated in Table 2. We can also infer that the model becomes more calibrated when using SWAG and SWAGBM, as inferred by the lower negative log-likelihood (NLL) values in Table 2. Notably, SWAGBM demonstrates superior calibration compared to standard SWAG. Impulse Noise, one of the corruptions, showed a significant improvement with the SWAG-trained ViT model, rising from 68.26% to 74.40%, which further increased to 86.74 when using SWAGBM. Likewise, Glass Blur improved from 70.12% to 77.87%, demonstrating the advantages of using SWAGBM over SWAG. From Figure 12, we can conclude that larger models, such as ViT, exhibit overconfident predictions. **However, the SWAGBM models are effective in calibrating these predictions and can outperform multiple optimizers, including SGD and Adam.**

Table 1: Comparison of SGD, SWAG, and SWAGBM regarding the accuracy and NLL values of the PreActResNet-164 model using the ImageNet dataset.

| Noise Type | SGD | | SWAG | | SWAGBM | |
|---|---|---|---|---|---|---|
| | Accuracy (%) | NLL | Accuracy (%) | NLL | Accuracy (%) | NLL |
| Clean | 82.01 | 3.69 | 91.60 | 2.43 | **92.21** | **2.01** |
| Pixelate | 30.21 | 3.69 | 48.09 | 2.41 | **49.45** | **1.98** |
| Contrast | 4.67 | 6.34 | 17.49 | 5.11 | **20.05** | **4.98** |
| Elastic Transform | 18.26 | 5.40 | 46.67 | 2.61 | **50.40** | **2.12** |
| JPEG | 37.61 | 3.15 | 40.22 | 2.11 | **41.56** | **1.76** |
| Snow | 17.65 | 4.97 | 38.83 | 3.15 | **39.23** | **2.41** |
| Fog | 25.40 | 4.14 | 49.36 | 2.43 | **51.67** | **1.97** |
| Frost | 22.82 | 4.55 | 36.28 | 3.42 | **39.45** | **2.67** |
| Brightness | 56.71 | 1.89 | 66.79 | 1.79 | **70.22** | **1.65** |

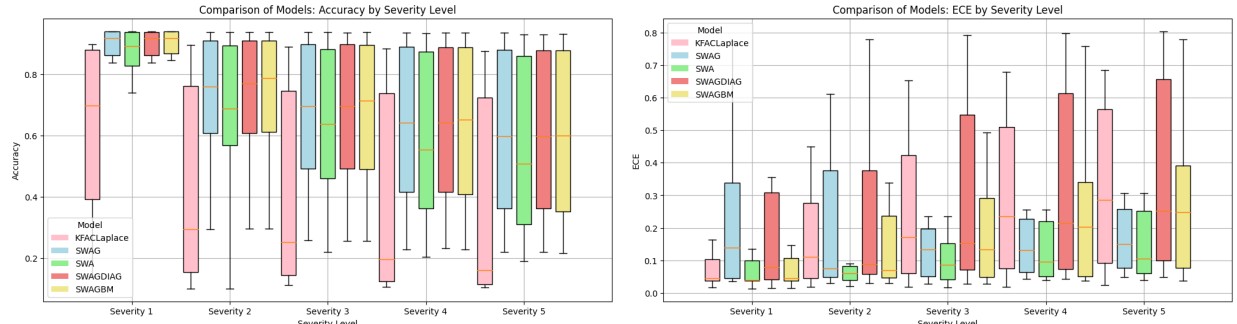

Figure 9: Box plots of accuracy and ECE values for CIFAR-10 images under the VGG16 model across various severity levels.

Table 2: Comparison of ADAM, SWAG, and SWAGBM in terms of accuracy and NLL values on CIFAR-100 with ViT model.

| Noise Type | ADAM | | SWAG | | SWAGBM | |
|---|---|---|---|---|---|---|
| | Accuracy (%) | NLL | Accuracy (%) | NLL | Accuracy (%) | NLL |
| Clean | 97.02 | 0.36 | 98.73 | 0.21 | **98.90** | **0.19** |
| Brightness | 95.00 | 0.24 | 96.73 | 0.11 | **98.27** | **0.06** |
| Defocus Blur | 91.84 | 0.26 | 92.38 | 0.25 | **96.96** | **0.11** |
| Elastic Transform | 78.44 | 0.76 | 80.07 | 0.71 | **92.82** | **0.27** |
| Fog | 84.45 | 0.51 | 86.50 | 0.46 | **95.48** | **0.16** |
| Frost | 91.01 | 0.30 | 92.56 | 0.33 | **95.23** | **0.17** |
| Snow | 92.70 | 0.23 | 92.87 | 0.23 | **95.78** | **0.15** |
| Gaussian Blur | 90.34 | 0.31 | 91.52 | 0.28 | **96.37** | **0.13** |
| Glass Blur | 67.45 | 1.08 | 70.12 | 1.03 | **77.87** | **0.81** |
| Impulse Noise | 68.26 | 1.07 | 74.40 | 0.80 | **86.74** | **0.49** |
| JPEG | 85.81 | 0.46 | 85.88 | 0.47 | **91.77** | **0.30** |
| Saturate | 96.11 | 0.12 | 96.42 | 0.12 | **97.42** | **0.09** |
| Spatter | 93.65 | 0.22 | 95.02 | 0.17 | **96.66** | **0.11** |
| Zoom Blur | 91.88 | 0.25 | 92.87 | 0.24 | **96.15** | **0.14** |

## 4.6 Adversarial Attacks and Beyond

To further extend our investigation, we conducted additional experiments using various adversarial attacks on the CIFAR-10 dataset under the VGG-16 model. Here, we provide brief findings against each adversarial attack to reflect that the SWAGBM and SWAG models can better handle them than traditional SGD or ADAM-trained models. Notably, in white-box attacks Goodfellow et al. (2015)—where the adversary has knowledge of the model and leverages gradient-based methods—the SWAG AND SWAGBM models demonstrated improved resistance, effectively mitigating the impact of such attacks. (i) Under the FGSM (Fast Gradient Sign Method) attack Goodfellow et al. (2015) with $\epsilon = 0.004$, the SGD-trained model achieved an accuracy of 13.28%, while the SWAG-trained model improved this to 30.24%. Notably, the SWAGBM variant further increased accuracy to 36.12%. As shown in Table 4, the lower NLL values for SWAGBM confirm that its calibration is superior, enhancing its robustness against adversarial perturbations.

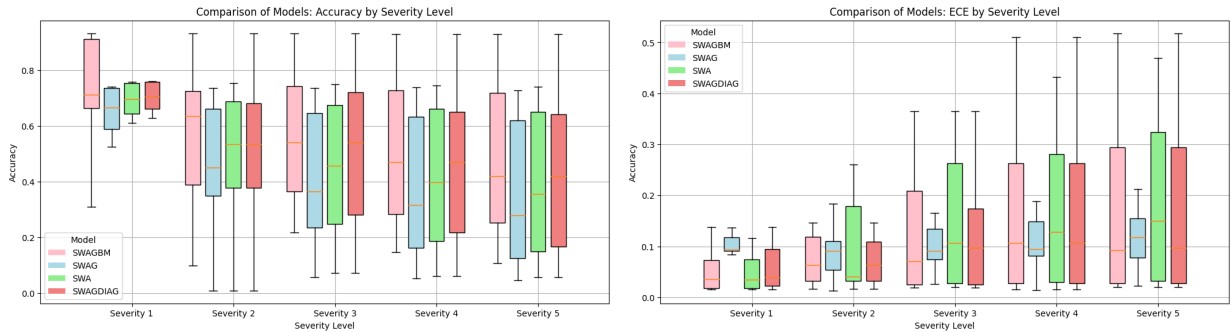

Figure 10: Box plots of accuracy and ECE values for CIFAR-100 images under the VGG16 model across various severity levels.

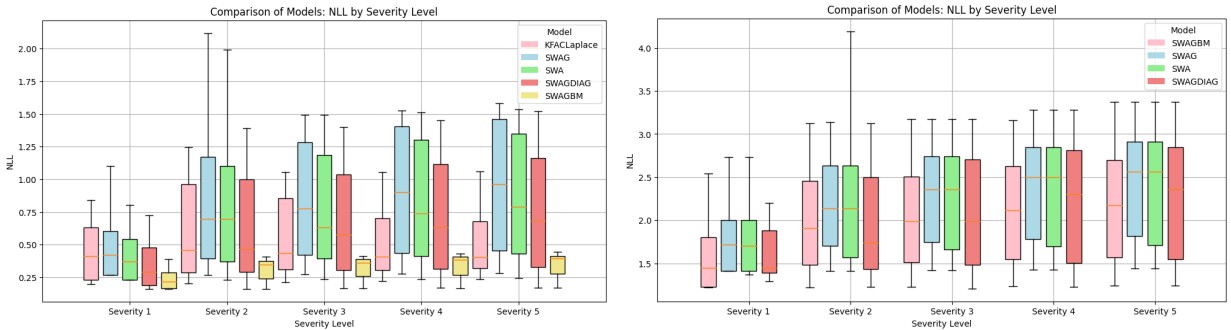

Figure 11: Box plots of NLL values for CIFAR-10 and CIFAR-100 dataset under the VGG16 model across various severity levels.

(ii) Under the more challenging PGD (Projected Gradient Descent) attack Madry (2017) with a maximum perturbation $\epsilon = 0.03$, a step size of 0.01, and 50 iterations, the SWAG-trained model achieved an accuracy of 10.28% compared to only 3.24% for the SGD-trained model (see Table 3). Although SWAGBM exhibited higher accuracy at lower perturbation levels that gradually decreased compared to standard SWAG, its consistently lower NLL values demonstrate that SWAGBM enhances adversarial robustness overall. We also experimented with the Carlini & Wagner (C&W) attack Carlini & Wagner (2017), the SWAG-trained model attained an accuracy of 78%, outperforming the SGD-trained model's 71%. This attack was configured with a confidence parameter of 10, a learning rate of 0.01, and 1,000 iterations.

Table 3: Accuracy and NLL under PGD attack on CIFAR-10 using VGG16 model.

| $\epsilon$ | SGD | SWAG | | SWAG BM | |
|---|---|---|---|---|---|
| | Accuracy | Accuracy | NLL | Accuracy | NLL |
| 1/255 | 23.21% | 35.68% | 0.33 | **39.29%** | **0.91** |
| 2/255 | 15.50% | **28.50%** | 0.40 | 28.14% | **0.25** |
| 3/255 | 10.89% | 22.74% | 0.45 | **22.8%** | **0.32** |
| 4/255 | 7.08% | **18.08%** | 0.53 | 14.89% | **0.39** |
| 5/255 | 6.37% | **15.56%** | 0.59 | 11.74% | **0.46** |
| 6/255 | 5.12% | **12.67%** | 0.65 | 9.40% | **0.53** |
| 7/255 | 3.54% | **11.87%** | 0.71 | 7.62% | **0.59** |
| 8/255 | 3.24% | **10.28%** | 0.76 | 6.41% | **0.66** |

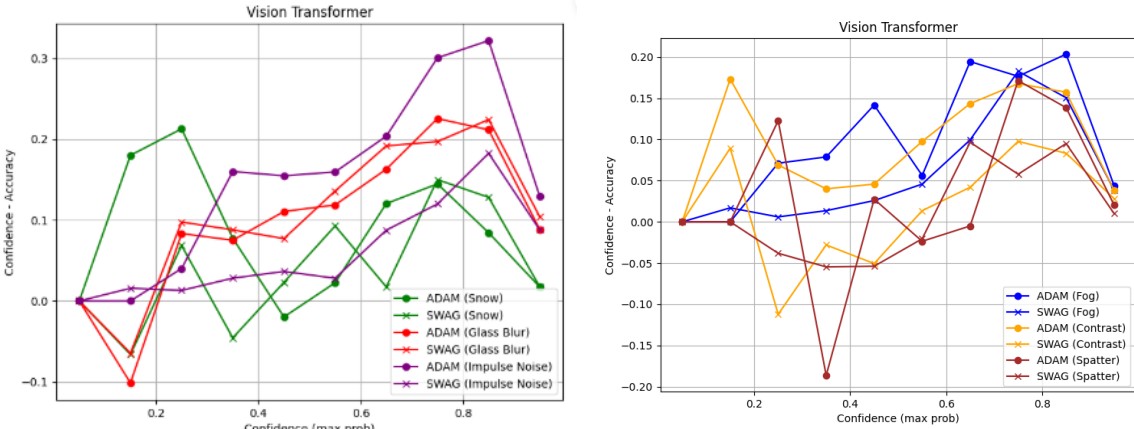

Figure 12: Box plots of accuracy and ECE values for CIFAR-10 images under the VGG16 model across various severity levels.

Table 4: Accuracy and NLL under FGSM attack on CIFAR-10 using VGG16 model.

| $\epsilon$ | SGD | SWAG | | SWAGBM | |
|---|---|---|---|---|---|
| | Accuracy | Accuracy | NLL | Accuracy | NLL |
| 1/255 | 13.78% | 30.24% | 0.36 | **36.12%** | **0.24** |
| 2/255 | 13.24% | 29.89% | 0.36 | **34.21%** | **0.25** |
| 3/255 | 13.11% | 27.24% | 0.37 | **32.45%** | **0.25** |
| 4/255 | 12.29% | 26.89% | 0.38 | **31.12%** | **0.26** |
| 5/255 | 11.56% | 24.36% | 0.35 | **30.11%** | **0.26** |
| 6/255 | 11.21% | 23.15% | 0.36 | **29.12%** | **0.23** |
| 7/255 | 10.43% | 22.19% | 0.39 | **28.00%** | **0.27** |
| 8/255 | 10.17% | 21.24% | 0.34 | **28.41%** | **0.28** |

## 5  Conclusion and Future Work

Interestingly, after learning about the deep network's vulnerability against corruption, a race against developing 'new' robust models has started. However, to tackle this issue and understand why the existing models are not strong about natural corruption, we worked in the direction of model calibration. After conducting a detailed analysis and extensive experimentation, we found that increased calibration leads to better robustness and performance on corrupted data. The reliability diagrams illustrate that, in natural noise, CNNs and ViT trained with standard methods become excessively overconfident in their predictions. Conversely, when teaching the models using Stochastic Weight Averaging Gaussian and Stochastic Weight Averaging Gaussian with Batch Means, we observed that the confidence scores aligned more with actual performance, leading to better-calibrated and robust predictions. Thus, for real-world deployment scenarios, it is crucial to consider training with a strategy that can better calibrate the model in its prediction since the world is inherently noisy (Pedraza et al., 2022; Chen et al., 2023) and every time developing a new robust model, leaving a non-robust model behind can lead to a hazardous solution.

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
