# OpenReview forum: "Trust Me, I’m Calibrated: Robustifying Deep Networks"
_TMLR — Rejected by TMLR_

### Review · Reviewer_2Ruc · 2025-05-10

**Summary Of Contributions:**

# Summary of Contributions

This paper proposes to evaluate the calibration degree of deep neural networks under different environmental effects and adversarial attack, which have been proposed by previous work for out-of-distribution detection. It modifies an existing Bayesian model to quantify the prediction uncertainty and calibrate DNN models. The proposed method is evaluated on CIFAR datasets for both CNN-based and transformer-based DNN models. The proposed method is compared with the baseline model, upon which the proposed modification is incorporated.

**Audience:**

No

**Broader Impact Concerns:**

N.A.

**Claims And Evidence:**

No

**Requested Changes:**

# Requested Changes

1. Present significant technical contribution and justify the proposed modification.

2. Include comprehensive experiments and comparison with existing methods on both experimental settings.

3. Include details for evaluation metrics and justify the experimental settings.

**Strengths And Weaknesses:**

# Weakness

1. **Trivial Technical Contribution**: This work proposes a trivial modification on an existing method - SWAG by collecting model weights after batches as opposed to after epochs proposed in the original work. The motivation of the proposed modification, as claimed in  statement (Section 3.2) quoted below, is unverified / questionable. First, reduced variance in covariance estimation does not always result in more accurate posterior approximation. Secondly, the approximated posterior weight distribution is for model trained on clean (un-corrupted) images, which lacks a connection or generalisation proof to guarantee improved calibration under a distributional shift from clean data to corrupted data.
> This additional averaging step reduces the variance in the covariance estimates, leading to a smoother and more accurate posterior approximation. As a result, this approach yields models with improved calibration and enhanced robustness, particularly in environments with noisy or adversarial data.

2. **Trivial Experiments**: In both experiments on model calibration and robustness, the proposed method is compared with only baseline methods, which are (i) model optimised with vanilla empirical risk minimisation and (ii) SWAG model. None of the existing model calibration methods and adversarial robustness methods are included for comparison.

3. **Unclear Evaluation Metrics**: Expected calibration error (ECE) is usually approximated with bin-based methods. However, there are numerous bin-based approximation methods available, thus it is necessary to include the details of the used metrics. As shown in Figure 1 and 2 in the paper, it is obvious that ECE approximation methods with different binning scheme and hyperparameters have been used for different datasets and models, with no details given.

---

### Review · Reviewer_6SbA · 2025-05-14

**Summary Of Contributions:**

This submission studies how calibrated neural networks, especially those based on Stochastic Weight Averaging-Gaussian (SWAG), help to improve the robustness against natural corruptions. An improvement of SWAG, namely SWAGBM, is further proposed to improve the calibration and robustness through more stable parameter estimation. Experiments are conducted on CIFAR-10, ImageNet, and their corrupted versions on both CNNs and ViTs.

**Audience:**

Yes

**Broader Impact Concerns:**

No concern in this regard.

**Claims And Evidence:**

No

**Requested Changes:**

There are several minor writing issues:
1. Section 1 "multiple benchmark object recognition datasets" is gramatically strange -> "multiple datasets/benchmarks on object recognition".

2. Section 3.2, cross reference "[1]" should be "Algorithm 1"

3. Secion 4.4 "Figures9" -> "Figures 9"

4. Section 4.6 "SWAG AND SWAGBM" -> "SWAG and SWAGBM"

5. Section 4.6 "We also experimented with the Carlini & Wager attack, the SWAG-trained model ..." two halves are both independent sentences.

6. Section 4.2, "Table ??" missing reference

**Strengths And Weaknesses:**

Strengths:
- The study of model calibration in the context of robustness against corrupted noises is interesting, and would be of interest to practitioners as calibration and natural curruptions are concerning topics in industry.

- It is remarkable to see how calibrated models greatly improve both robustness and calibration at the same time on various settings.

Weaknesses:
- The technical contributions are incremental. SWAGBM is proposed to just slightly improve SWAG by accumulating at the batch level.

- On the study side, though the title and abstract illustrate calibrated DNNs in general, the study scope is limited to only the SWAG approach. The paper needs to analyze other calibration approaches and their impacts on robustness and calibration.

- Writing quality can be greatly improved. A few important details are missing. For example:
1. A more detailed illustration of SWAG is needed. The explanation in Section 3.1 is a bit rough.
2. The algorithm 1 may need more comments from Line 16 to Line 19. Where does the improvement with Batch Means appear?
3. How is the covariance matrix computed? Some explicit formula should help.
4. Please define Expected Calibration Error (computation formula missing).

- Experiment evaluation can be further enhanced.
1. What is the inference overhead of SWAG and SWAGBM in practice compared to the standard SGD?
2. Why does SWAGBM disappear in Figures 3,4,5,6?

---

### Review · Reviewer_jQeA · 2025-06-02

**Summary Of Contributions:**

The authors present an experimental study on the effect of an approximate Bayesian inference method from previous work, SWAG, on robustness.
Furthermore, the authors propose to add an averaging step for the collection of the model checkpoints to use for SWAG, resulting in what the authors call SWAGBM.
The provided results show that SWAG and SWAGBM have a positive effect on robustness and calibration (mainly in the context of natural OOD shifts, with a smaller set of experiments on an adversarial setting) compared to approaches not based on Bayesian inference (SGD, SWA, Adam).

**Audience:**

No

**Broader Impact Concerns:**

None.

**Claims And Evidence:**

No

**Requested Changes:**

Most importantly, the authors should contrast their findings with those from the studies noted in the "Strengths and Weaknesses" section, clearly stating what is the additional contribution associated with the submission.

The authors should also more carefully support the empirical utility of SWAGBM compared to standard SWAG (see Strengths and Weaknesses" section), and provide more baselines on both adversarial robustness and OOD generalization.

**Strengths And Weaknesses:**

The idea of exploring the interaction between calibration and robustness could definitely be interesting, but the results shown in this submission would appear to be fairly well-known in the community. This is the main reason behind my "no" on both the claims and audience questions.
- The authors claim that "*no existing study understanding the correlation between confidence and robustness*" exists, using this as one of the main motivations for the work, but at the very least this was analyzed in the RobustBench paper [1], and within [2].
- The submission fails to cite a closely-related empirical study of SWAG on OOD contexts from NeurIPS 2023 [3].
- Beyond SWAG, the relative robustness of BNNs is well-known in the area, see for instance [4].

Specific remarks follow:
- The utility of SWAGBM over SWAG is not immediate. The authors claim that its uncertainty estimates are less extreme, but Figure 7 has a different color code for the two plots, with SWAGBM attaining a larger maximum entry. For instance, the improvements are not clear in Figure 2 (CIFAR-100) and in table 3 either, the latter showing that SWAG appears to be preferable under stronger attacks.
- The algorithmic contribution of SWAGBM is extremely minimal (not necessarily a problem on its own, but definitely one in the overall context of the submission).
- Adversarial training schemes should be discussed and compared against, for both OOD, where they are known to help [5], and adversarial settings. More baselines on OOD robustness are also desirable.
- The quality of the writing can be improved: text is repeated at the end of page 2, for instance, and some acronyms are defined multiple times (SWAG).
- The quality of the figures can be improved too: the resolution of some of them appears to be low (Figure 12), for instance.
- The description of the experimental evaluation starts by commenting on Figure 7: perhaps this could be placed earlier. It would also be nice to have consistent colors for each method across plots.


### References

[1] RobustBench: a standardized adversarial robustness benchmark, Croce et al., NeurIPS 2021

[2] Improving Calibration through the Relationship with Adversarial Robustness, Qin et al., NeurIPS 2021

[3] Beyond Deep Ensembles: A Large-Scale Evaluation of Bayesian Deep Learning under Distribution Shift, NeurIPS 2023

[4] Robustness of Bayesian Neural Networks to Gradient-Based Attacks, Carbone et al., NeurIPS 2020

[5] Improved OOD Generalization via Adversarial Training and Pre-training, Yi et al., ICML 2021

---

### Decision · Action_Editor_k3DK · 2025-06-24

**Recommendation:** Reject

**Audience:**

Yes

**Audience Explanation:**

Those who work in adversarial robustness and model calibration will be interested.

**Claims And Evidence:**

No

**Claims Explanation:**

All three reviewers raise consistent and serious concerns regarding the technical novelty, empirical rigor, and clarity of presentation. The core contribution—SWAGBM—constitutes only a minor modification to an existing method (SWAG), and its benefits over SWAG remain unconvincing due to weak empirical evidence and limited analysis. Furthermore, the experimental evaluation lacks comparison with strong baselines in both calibration and robustness, and omits recent related work that undermines the claimed novelty. Reviewers also point to critical issues in writing quality, missing methodological details, and inconsistencies in evaluation metrics.

Most importantly, the authors did not provide a rebuttal to address these substantial concerns. Given the triviality of the proposed method, insufficient experimental validation, and absence of engagement with reviewer feedback, the paper could not be considered for publication in TMLR.